# Deep Submarine Landslide Contribution to the 2010 Haiti Earthquake Tsunami

Adrien Poupardin[1,2], Eric Calais[2], Philippe Heinrich[3], Hélène Hébert[3], Mathieu Rodriguez[2], Sylvie Leroy[4], Hideo Aochi[2,5], and Roby Douilly [6]

[1]Institut de Recherche en Constructibilité, ESTP, Université Paris Est, Champ-sur-Marne, 77420, France
[2]Ecole normale supérieure, Dept. of Geosciences, University PSL, CNRS, Paris, 75005, France
[3]Commissariat à l'Energie Atomique, DAM, DIF, Arpajon, 91290, France
[4]Université Pierre et Marie Curie, Sorbonne Universités, CNRS, ISTeP, Paris, 75005, France
[5]Bureau de Recherches Géologiques et Minières, Orléans, 45000, France
[6]University of California, Riverside, Department of Earth Sciences, Riverside, 231, USA

*Correspondence to*: Adrien Poupardin (apoupardin@estp-paris.eu)

**Abstract.** The devastating $M_w$ 7.1, 2010, Haiti earthquake was accompanied by local tsunamis that caused fatalities and damage to coastal infrastructure. Some were triggered by slope failures of river deltas in close vicinity of the epicenter, while others, 30 to 50 km to the north across the Bay of Gonâve, are well explained by the reverse component of coseismic ground motion that accompanied this mostly strike-slip event. However, observations of run-up heights up to 2 m along the southern coast of the island at distances up to 100 km from the epicenter, as well as tide gauge and DART buoy records at distances up to 600 km from the epicenter have not yet received an explanation. Here we demonstrate that these observations require a secondary source, most likely a submarine landslide. We identify a landslide scar 30 km from the epicenter off the southern coast of Haiti at a depth of 3500 m, where ground acceleration would have been sufficient to trigger slope failure in soft sediments. This candidate source, 2 km$^3$ in volume, matches observation remarkably well assuming that the sediment collapse obeys a viscous flow with an initial apparent viscosity of $2\times10^5$ Pa.s. Although that particular source cannot be proven to have been activated in 2010, our results add to a line of evidence that earthquake-triggered submarine landslides can cause significant tsunamis in areas of strike-slip tectonic regime.

## 1 Introduction

The devastating $M_w$ 7.1, 2010, Haiti earthquake occurred within the Caribbean-North America plate boundary, where oblique relative motion at 19 mm/yr is partitioned between shortening on the North Hispaniola fault to the north and strike-slip motion on the E–W striking Septentrional and Enriquillo faults throughout the island (Fig. 1; Symithe et al., 2015). Although the earthquake was first assumed to have occurred on the Enriquillo fault, several independent studies later showed that it actually ruptured a previously unmapped fault – the Léogâne fault – with a source mechanism combining strike-slip and reverse faulting (Calais et al., 2010; Hayes et al., 2010; Hashimoto et al., 2011; Symithe et al., 2013), in a setting resembling the 1989 Loma Prieta earthquake in California (e.g., Dietz and Ellsworth, 1990).

In spite of its moderate size and primarily strike-slip mechanism, the earthquake was accompanied by local tsunamis that caused at least three fatalities in Grand Goâve (Fritz et al., 2013) and some damage to coastal infrastructure. Thanks to the rapid deployment of an International Tsunami Survey Team (ITST), run-up height, flow depth, and inundation were documented at more than 20 sites along Haitian coastline (Fig. 1; Fritz et al., 2013). Maximum tsunami heights reached 3 m in the close vicinity of the earthquake epicenter in the bays of Petit Goâve and Grand Goâve along the northern coast of the Southern Peninsula, where the tsunami was triggered by slope failures of river deltas (Hornbach et al., 2010; Fritz et al., 2013). Thirty to fifty km to the north across the Bay of Gonâve, eyewitnesses reported drawdown near the locality of Luly, well explained by the reverse component of the coseismic ground motion (Hornbach et al., 2010).

However, observations of run-up heights up to two meters along the southern coast of Haiti, at distances up to 100 km from the epicenter, as well as tide gauge and DART buoy records at distances of 300 km and 600 km from the epicenter, respectively, have not yet received a satisfactory explanation. Fritz et al. (2013), using the early source model from the National Earthquake Information Center (NEIC), were able to roughly match these observations by scaling coseismic slip by a factor of four while keeping a constant seismic moment, hence downscaling the regional rigidity coefficient. A similar *ad hoc* procedure was also proposed by Newman et al. (2011) for the 2010 Mentawai, Indonesia, earthquake in order to account for the discrepancy between coseismic slip inverted from teleseismic waves and actual slip derived from geodetic measurements. This artificially lowered rigidity can indeed account for slow ruptures at subductions – "tsunami earthquakes" (Kanamori, 1972) – such as in Mentawai in 2010 or in Java in 2006 (Hébert et al., 2012), but the Haiti earthquake fault ruptured at 2.6 km/s, a regular rupture velocity (de Lépinay et al., 2011).

Here we revisit this issue by (1) testing most of the finite-source models proposed for the 2010 Haiti earthquake, (2) investigating the role that an earthquake-triggered landslide off the southern coast of Haiti may have played in tsunami generation. We show that this latter hypothesis is very likely, in line with the growing set of evidence linking tsunamis along strike-slip faults to submarine landslides (e.g., Ma et al., 1991; Yalçıner et al., 2002; Rodriguez et al., 2017).

## 2 Tsunami Calculation Method

The numerical method involves modeling of the initiation, propagation, and run-up of the tsunami waves. For the earthquake source, we calculate the initial seafloor perturbation – assumed to be instantaneously and fully transmitted to the water column – using coseismic static ground displacement for a dislocation in an elastic half-space (Okada, 1992). For the landslide source, we assumed that the sediment collapse obeys a viscous flow, consistent with the mud or shale nature of the sediments involved (Assier-Rzadkiewicz et al., 2000). We keep the dynamic viscosity constant during the whole simulation, which may not be realistic in terms of landslide propagation but has little effect on the free surface, which is mostly sensitive to the initial landslide acceleration (e.g., Løvholt et al, 2015; Poupardin et al., 2017).

The tsunami propagation calculation accounts for dispersive terms in the deep ocean (Boussinesq equations) and neglects them near the coast (St Venant equations), retaining only the lowest-order nonlinear terms (Poupardin et al., 2018). In the latter case, we use a shock-capturing method to propagate strong nonlinear waves. In the former one, we use an iterative Crank-Nicholson

method to solve the Boussinesq equations (details in Poupardin et al., 2017). Propagation equations are solved using a finite-difference scheme with a spatial discretization that uses centered differences for linear terms and forward differences for advection terms (Hébert et al., 2001).

We solve the numerical problem using four successive levels of nested bathymetric grids from the deep ocean to the local coastal areas of interest. The two lower resolution grids use the GEBCO World Bathymetry with a resolution from 1600 to
400 m. We digitized local bathymetric charts and included existing digital bathymetric data in the bays of Port-au-Prince, Jacmel, and Santo-Domingo to build local grids with a resolution of 25 m in order to accurately model near-shore resonance and amplification. We used the 1600 m resolution grid to propagate the tsunami to the DART Buoy and the 25 m one for the Santo Domingo tide gauge in order to account for resonance due to coastal bathymetry.

## 3 Earthquake-generated tsunami

We tested a uniform slip of 1 m using the NEIC fault geometry and rupture mechanism, as in Fritz et al. (2013; hereafter denoted the F model), as well as four finite-source models with variable coseismic slip derived from combinations of GPS, InSAR, coastal uplift, and teleseismic data (Hayes et al., 2010; Meng et al., 2012; Symithe et al., 2013; Saint Fleur et al., 2015; hereafter denoted the H, M, S, and SF models, respectively). All finite-source models show that rupture occurred on a north-dipping blind fault (Leogâne fault) with 2/3 of the moment released by strike-slip motion and 1/3 by reverse motion, consistent
with up to 60 cm of coastal uplift observed in the epicentral area (Hayes et al., 2010). The NEIC source model, based on teleseismic data only, shows the same coseismic motion partitioning, but on a south-dipping fault with a north-verging reverse component.

Static coseismic displacements from the finite-source models show up to 0.5-1 m of seafloor uplift north of the rupture (Fig. 2) and less than 0.01 m of subsidence along the southern coastline. The F model shows an opposite pattern because of its fault
dip opposite to the finite-source models, and larger coseismic displacement because of the *ad hoc* coseismic slip scaling factor applied (see above). This model is inconsistent with the coseismic geodetic observations and the coseismic coastal uplift observed in the epicentral area. It matches observations at the Santo Domingo tide gauge and DART buoy 42407 reasonably well in amplitude and period at low frequencies, but its coseismic slip was arbitrarily tuned to match these observations (Fig. 3).
All finite-source models predict a small tsunami in the epicentral area, with the M and SF models predicting wave heights of up to 1 m along more than 10 km of coastline length, inconsistent with the observations reported by the ITST team (Fritz et al., 2013). Indeed, coseismic coastal uplift in the M and SF models are larger and more spread out than the observations reported in Hayes et al. (2010). All finite-source models predict a local tsunami at Luly, with 0.5-1 m run-up height (Fig. 2), consistent with observations at that location (Hornbach et al., 2010). This local tsunami likely occurred as a result of the
intersection of the coastline with the sharp edge of the bathymetric low that occupies most of the Bay of Port-au-Prince. Only the F model predicts significant wave heights along the southern coast (Fig. 4), but it also predicts run-up heights of 0.5-1 m

in the highly populated and low-lying coastal region surrounding the capital city of Port-au-Prince that were not observed after the earthquake (Fig. 2).

That none of the finite-source models comes close to the observations in the near field in Jacmel and Pedernales or in the far field at two locations 450 km apart likely indicates the presence of a secondary tsunami source. An offshore landslide comes to mind, as they are well known to trigger tsunamis even at large distances (e.g., Heinrich et al., 2001; Okal and Synolakis, 2004; Fritz et al., 2007; Hébert et al., 2012). Fritz el al. (2013) indeed already suspected that this type of source may be required to explain field observations near Jacmel and Pedernales.

## 4 Landslide-generated tsunami

We use a high-resolution bathymetry recently acquired offshore the southern coast of Hispaniola (Leroy et al., 2015) to search for scars in the seafloor morphology that could be indicative of large submarine landslides within the area significantly shaken by the earthquake. The fact that the earthquake source approximately matches observed arrival times at the tide gauge and DART buoy is an indication that a possible landslide source should be located fairly close to the epicentral area, which is a requirement for seafloor acceleration to be sufficient to trigger slope failure.

We find the clearest and largest landslide signature 30 km due south of the tip of the cape that bounds the bay of Jacmel to the west, at a water depth of 3500 m (Fig. 5). The landslide is expressed on the seafloor by two arcuate scars at the slope break between a flat plateau (Haiti Plateau) marked by a sediment wave field and a narrow canyon likely feeding the northeastern corner of the 5000 m-deep Haiti sub-basin (Mauffret and Leroy, 1999). The scar shows two horseshoe-shaped lobes with sharp edges, opening to the east onto the narrow canyon that likely acted as an evacuation pathway for the sediments. Unfortunately, no chirp or high-resolution seismic data is available crossing that feature, which has not been cored either, so that one cannot determine the age of the sediment collapse. We try to identify the bathymetric scar using GEBCO grids, which contains data older than the 2010 Haiti earthquake, but we could not find any evidence for it. This could mean that the scarp is post-earthquake, but equally well that the resolution of the GEBCO data is insufficient to detect such a small-scale feature.

We test that landslide as the plausible secondary source necessary to explain the observations in the near field (southern coastline) and far field (tide gauge and DART buoy) of the earthquake. We use the detailed bathymetry (Fig. 5) to reconstruct the sedimentary volume involved in the landslide, as in ten Brink et al. (2006). It consists of filling-in the failed area according to the adjacent scar height, to reconstruct the seafloor morphology prior to the slide. By using this method, we find a volume of 2 km$^3$, a maximum thickness of 150 m, and a slope of 10°. Once triggered, the failed volume flows with maximum velocities in the 10–65 m/s range depending on the assumed viscosity (Fig. 6). The initial acceleration remains approximately the same at ~0.5 m/s² for all tested dynamic viscosities but it quickly decreases for the dynamic viscosity of $2\times10^5$ Pa.s, resulting in smaller amplitudes within the model domain. The influence of the viscosity is evaluated by considering water heights calculated at a synthetic gauge located offshore, a few kilometer north of the landslide (gauge $G_1$ in Fig. 8). As expected, the larger the viscosity, the smaller the amplitudes of water waves, whereas periods are unchanged (Fig. 7). The largest viscosity results in maximum wave amplitudes below 50 cm at the location of gauge $G_1$.

In our preferred model, the southern coast of Haiti is reached by water waves within 5 to 10 minutes of the landslide triggering. Maximum water heights after one hour of simulation (Fig. 8) show that water waves are amplified along an arc segment extending from the landslide area towards the Bay of Jacmel. The maximum wave heights reach 2 to 3 m along a coastal segment of 10 km for a viscosity of $2\times10^5$ Pa.s (Fig. 8) consistent with ITST observations (Fritz et al., 2013).

Model results at synthetic gauge $G_2$ located in the bay of Jacmel show that maximum wave amplitudes range from 3 to 6 m

depending on the viscosity of the sliding material (Fig. 9). ITST measurements at Jacmel (see Fig. 5c in Fritz et al., 2013), around 3 m, favor a sliding material with a viscosity in the high range of the ones tested here. Due to the local amplification of the bay, the maximum water height at Jacmel occurs 11 minutes after the underwater slope failure consistent with observations.

Further along the coast, the comparison of calculated and observed time series at the Santo Domingo tide gauge is shown in

Fig. 10. Data are available with a 1-minute sampling rate and the tide gauge is located at the end of a small harbor that is poorly described in the simulation. Nevertheless, the arrival time of the first wave (about 45 min) as well as the periods and amplitudes of water waves are approximately reproduced by the model. A time lag of 8 minutes exists between observations and simulations (Fig. 10), which could be explained by imprecisions of the tide-gage clock, a delay of the collapse triggering, or imprecisions of the simulation due to errors in bathymetric data.

In the offshore direction, wave propagation is subject to frequency dispersion, which the model takes into account by solving Boussinesq equations. In Fig. 10, we compare the calculated time series with recorded data at DART Buoy 42407 with a sampling rate of 1 min. Due to frequency dispersion, the highest water height in the calculated wave train is not the first one but the third, with an amplitude of 0.5 cm. The arrival times as well as amplitudes of the first two waves match the observed signal well. Nevertheless, the oscillations of the following waves (around 3 min) have periods that are shorter than the observed

ones (around 4 minutes), which could indicate that the simulated landslide volume is insufficient to explain the observations.

**5 Discussion**

As a first verification, we checked that the landslide tested here, located 70 km from the epicenter, satisfied the magnitude-distance relationships proposed by Salamon and Dimanna (2019). Then, we ~~first~~ asked whether ground acceleration during the 2010 Haiti earthquake would be sufficient to trigger slope failure at the location of the earthquake described above. We estimate

the resonance frequency of the landslide using $f = V_s/4H$ where $V_s$ is the shear-wave velocity and $H$ the landslide thickness (Parolai et al., 2002). For $V_s = 1000$ m/s and $H = 100$ m $f$ is 2.5 Hz, a high-enough frequency to justify using PGA as a proxy for the acceleration that determines slope failure. It is generally recognized that 0.1–0.2 g PGA is the threshold of stability for the triggering of landslides in benthic sediments (e.g., Keefer, 1984, Meunier et al., 2007). Tanyaş et al. (2017) show that 80% of earthquake-induced terrestrial landslides are observed in the PGA interval 0.1-0.8 g. Offshore, sedimentation rates, the state

of sediment consolidation, and the occurrence of weak layers determine the potential of seismic shaking to increase pore pressure up to the failure threshold. As a result, large PGA does not always trigger mass flow, whereas low PGA can sometimes be sufficient (Hampton et al., 1996; Viesca and Rice, 2012; ten Brink et al., 2016; Pope et al., 2017). For example, the study

of Janin et al. (2019) on Holocene sediment collapses along the North Anatolian Fault in the Marmara Sea, shows that water circulation induced by seismic shaking in the material can have a significant effect on landslide triggering.

There was no direct measurement of regional ground acceleration during the 2010 Haiti earthquake, but indirect estimates from rigid body displacements and structural damage infer a PGA value of 0.2-0.4 g in Port-au-Prince (Olson et al., 2011; Goodno et al., 2011; Hough et al., 2012), ~30 km east of the earthquake epicenter. A numerical ground motion study based on the coseismic slip distribution from Hayes et al. (2010) finds the mean PGA in Port-au-Prince to be 0.20-0.33g (Mavroeidis and Scotti, 2013). We computed a first-order estimate of the horizontal ground acceleration at the location of the offshore

landslide described above using the "Next Generation Attenuation Ground Motions code" (Boore, 2012) for a M7 earthquake (Fig. 11). We find PGA values on the order of 0.1 g at a distance of 30 km to the earthquake epicenter on soft rock (shear wave velocity near surface Vs30 of 530 m/s). We also used the more precise, but more complex, approach based on a dynamic rupture simulation of the 2010 earthquake on a fault of realistic geometry, as described in Douilly et al. (2015, 2017). Using the same source as these authors, which satisfies observations of static coseismic ground displacement, we find the largest

horizontal peak ground acceleration at 0.18 g, for the E-W component while the mean horizontal peak ground acceleration is 0.17 g (Fig. 11). These two ground motion estimates are consistent with each other, though one must keep in mind that these calculations do not account for the more complex near-surface sediment structure likely to be present in the area of the submarine landslide.

Although there is no direct evidence that the submarine landslide described above was triggered by the 2010 Haiti earthquake,

it appears that seafloor acceleration in that area was within the range that permits slope failure in marine sediments. It is therefore plausible that a submarine landslide off the southern coast of Haiti is the secondary source necessary to explain, at the same time, the far-field tide gauge and DART buoy observations and the exceptionally large run-up heights at a few locations along the southern coast of Haiti given the limited coseismic displacement there.

A number of similar examples have been documented, often in subduction contexts. For instance, the 1998, $M_w$7.0, Papua

New Guinea earthquake was followed by a tsunami with run-up heights of over 7 m on a 20 km-long coastal segment, too large to be accounted for by the coseismic ground displacement alone (Okal and Synolakis, 2001; Tappin et al., 2008). Numerical simulations showed that the tsunami could be attributed to a submarine landslide with a volume of 4 km$^3$ located 20 km offshore (Heinrich et al., 2001). Similarly, the 2006 Java tsunami produced run-up heights up to 20 m at Permisan, indicative of a submarine landslide off the Nusa Kambangan Island (Fritz et al., 2007, Hébert et al., 2012). In the northern

Caribbean, López-Venegas et al. (2008) argue that the tsunami associated with the October 11, 1918 Mw7.5 normal faulting earthquake in the Mona Passage between Hispaniola and Puerto Rico was the result of a submarine landslide at a ~2000 m depth. Run-up heights in western Puerto Rico reached up to 6 m about 20 km from the landslide, whose volume was estimated at 10 km$^3$, five times that of the landslide described here. In a tectonic context similar to the 2010 Haiti earthquake, the 1989, $M_w$6.9, Loma Prieta earthquake excited tsunamis in the nearby Monterey Bay that required a secondary offshore landslide,

assumed to have occurred in shallow waters (Ma et al., 1991).

In northern Haiti, the Mw7.6 to 8.0, 1842 May 7 earthquake was followed by a tsunami that killed ~300 people, with wave heights of 2 m reported at a few locations along the northeastern Haitian coast and a maximum run-up of 4.6 m in the city of Port-de-Paix (Scherer, 1912). Some far-field effects have been reported, but their reliability remains questionable (O'Loughlin and Lander, 2003). Because it is the fastest-slipping active fault closest to the affected area, the Septentrional Fault is commonly assumed to be the source of that earthquake. However, a recent study shows that an 1842-like strike-slip earthquake on that fault cannot explain the tsunami run-up heights observed (Gailler et al., 2015). As a consequence, these authors suggest that event may have occurred on the North Hispaniola Fault, a slow-slipping reverse faults ~50 km offshore. The MMI IX intensities estimated along the Northern Haiti coastline for the 1842 earthquake suggest accelerations on the order of 0.3 g (Murphy and O'Brien 1977). It is therefore plausible that a strike-slip event on the Septentrional Fault would have triggered submarine slope failures – possibly multiple ones – along the steep and near-shore slopes that characterize the bathymetry of the northern Haiti margin (Leroy et al., 2015).

Underwater mass movements have been suggested for more than 100 years to trigger tsunamis in the near field of large earthquakes (e.g., Verbeek 1900; Gutenberg 1939). They are now considered to be significant contributors to tsunami hazard (e.g. Synolakis et al. 2002; ten Brink et al., 2006; Okal et al. 2009). It has been noted that tsunamis in strike-slip tectonic regimes were more frequent than expected. Such events may be accompanied by submarine or subaerial landslides (e.g. Ma et al., 1991; Imamura et al., 1995, Yalçıner et al., 2002; Hornbach et al., 2010; Hoffman et al., 2014) or associated to a large co-seismic deformation (Frucht et al., 2019). That local earthquakes can trigger shallow submarine slope failures is well known and was documented following the 2010 Haiti earthquake, in its epicentral area (Hornbach et al., 2010; Fritz et al., 2013). That deep submarine landslides can also produce significant tsunamis is less clear, though it was for instance proposed for the Puerto Rico, $M_L$7.5, 1918 earthquake and tsunami (López-Venegas et al., 2008).

tenBrink et al. (2020) recently proposed that the tsunami observations described here were caused by a doublet of dynamically triggered aftershocks with magnitudes of M6.8 and M6.5 located 85 km southwest of the epicenter, with reverse faulting on south-dipping faults. We first note that these magnitudes are significantly larger than the M5.4 and 5.1 quoted in the original publication describing these aftershocks (Fan and Shearer, 2016). We computed the co-seismic displacements that such large magnitude events close to the coast would have caused at GPS sites onshore and found that they would have values (up to 5 cm in their preferred model) and directions (northward) that are not observed in the GPS data (Calais et al., 2010). Finally, our analysis of the detailed bathymetry and recently acquired seismic reflection profiles at the location of the alleged south-dipping reverse fault (Leroy et al., 2015) indicate that they are actually north-dipping. We therefore argue that the explanation proposed by tenBrink et al. (2020) does not hold in the face of the available data.

## 6 Conclusion

We have shown that observations of run-up heights up to 2 m along the southern coast of the Haiti at distances up to 100 km from the epicenter, as well as tide gauge and DART buoy records at distances up to 600 km from the January 12, 2010,

epicenter require a secondary source, and can be explained by a submarine landslide. We have identified a landslide scar 30 km from the epicenter off the southern coast of Haiti at a depth of 3500 m, where ground acceleration would have been sufficient

to trigger slope failure in soft sediments. This candidate source, 2 km$^3$ in volume, matches observation remarkably well assuming that the sediment collapse obeys a viscous flow with an initial apparent viscosity of $2 \times 10^5$ Pa.s. Although that particular source cannot be proven to have been activated in 2010, our results add to a line of evidence that earthquake-triggered submarine landslides can cause significant tsunamis in areas of strike-slip tectonic regime. This result contributes to explaining why many tsunamis occur along strike-slip fault systems, especially when steep submarine slopes are within reach of

significant coseismic ground acceleration, as is the case along most strike-slip fault systems in the Caribbean.

**Acknowledgements**

This work was funded by the "Yves Rocard" Joint Laboratory between the Ecole normale supérieure, the Commissariat à l'Energie Atomique, the CNRS, and the Interreg Caraibes project PREST (European Union and FEDER). The authors are grateful to colleagues from the Institut de Physique du Globe for sharing their source model for the 2010 Haiti earthquake.

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

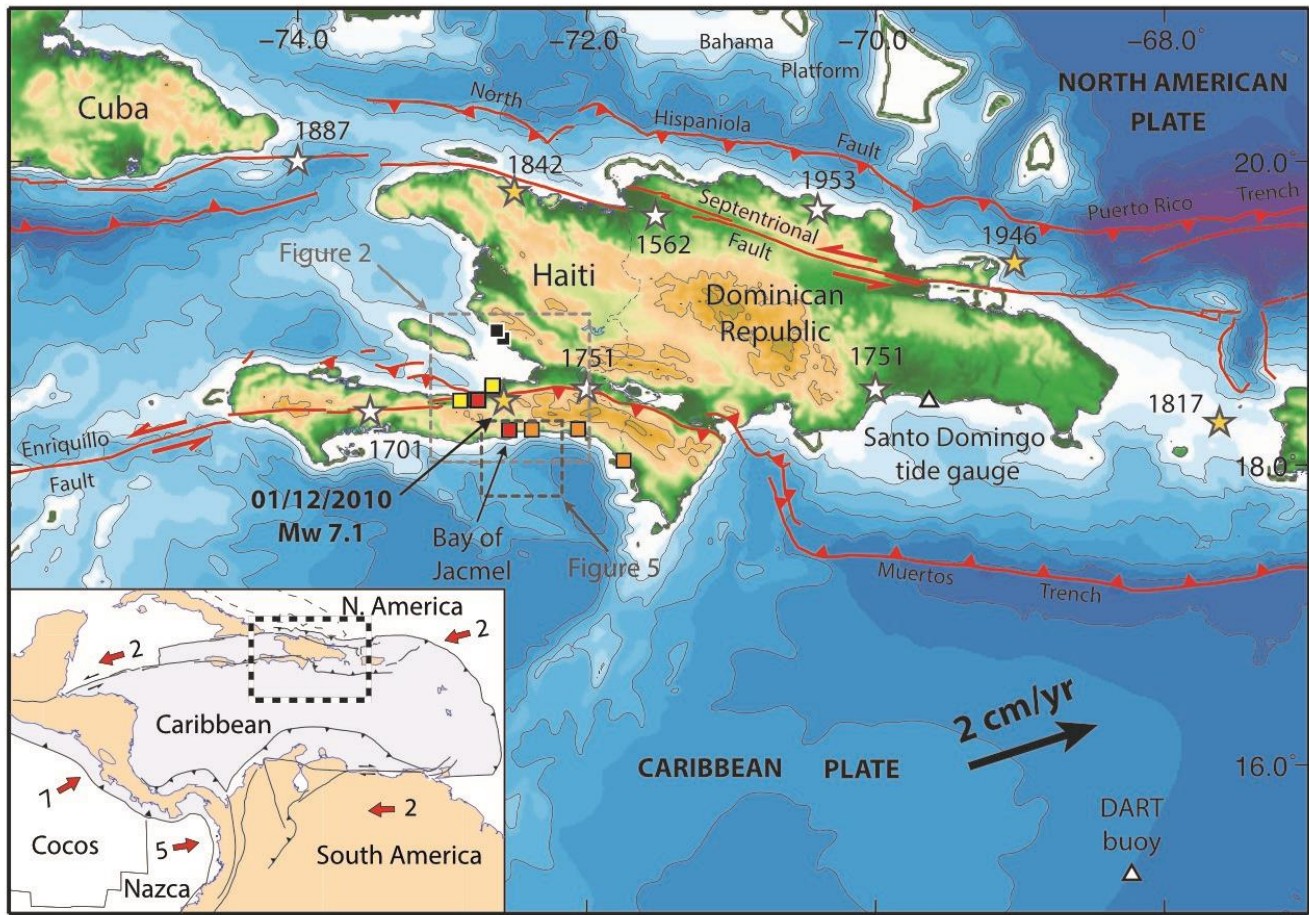

**Figure 1: Regional tectonic context. The yellow star indicates the epicenter of the 2010, $M_w$7.1, Haiti earthquake. Other yellow stars indicate the approximate epicenter of large tsunamigenic historical earthquakes, in particular the 1842 event of northern Haiti, which is discussed in the text. White stars show the approximate location of other large historical earthquakes. Tsunami run-up heights observed after the 2010 event are indicated by squares: < 1 m (yellow), 1 to 2 m (orange), > 2 m (red), draw down (black). The locations of DART buoy 42407 and Santo Domingo tide gauge are indicated by white triangles. Inset shows large-scale tectonic framework with current plate velocities relative to the Caribbean shown in cm/yr.**


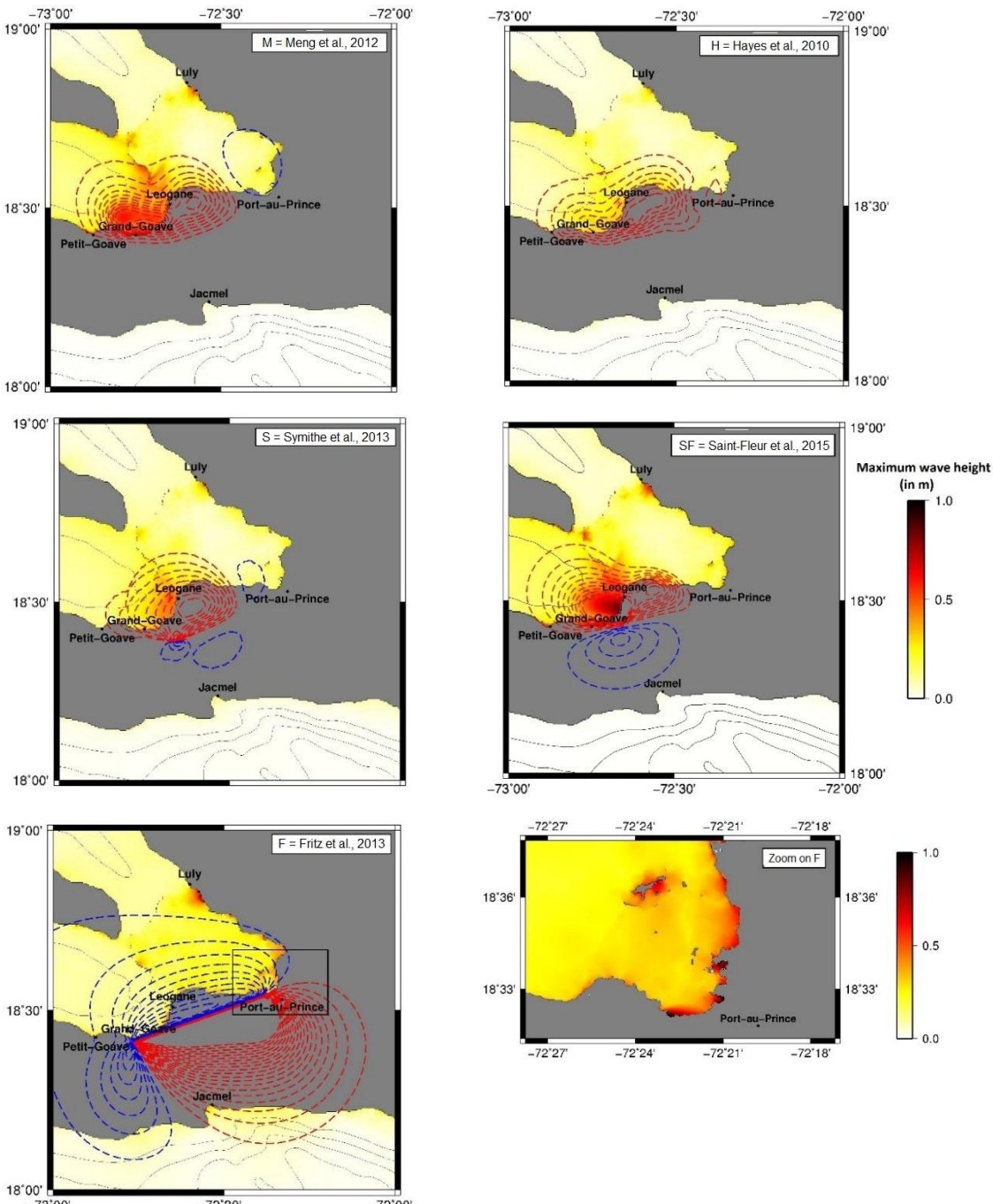

**Figure 2:** Initial coseismic seafloor displacement shown as dashed lines (red = uplift, blue = subsidence, contours are shown every 0.05 m) with maximum wave height shown as colored map in the background in the near field of the 2010 Haiti earthquake. H = Hayes et al., 2010, M = Meng et al., 2012, S = Symithe et al., 2013, SF = Saint Fleur et al., 2015, F =Fritz et al, 2013. Calculations are done on a 30' bathymetric grid resolution. Bottom right: blowup of the Port-au-Prince area for the F source model calculated using a 25 m resolution grid.

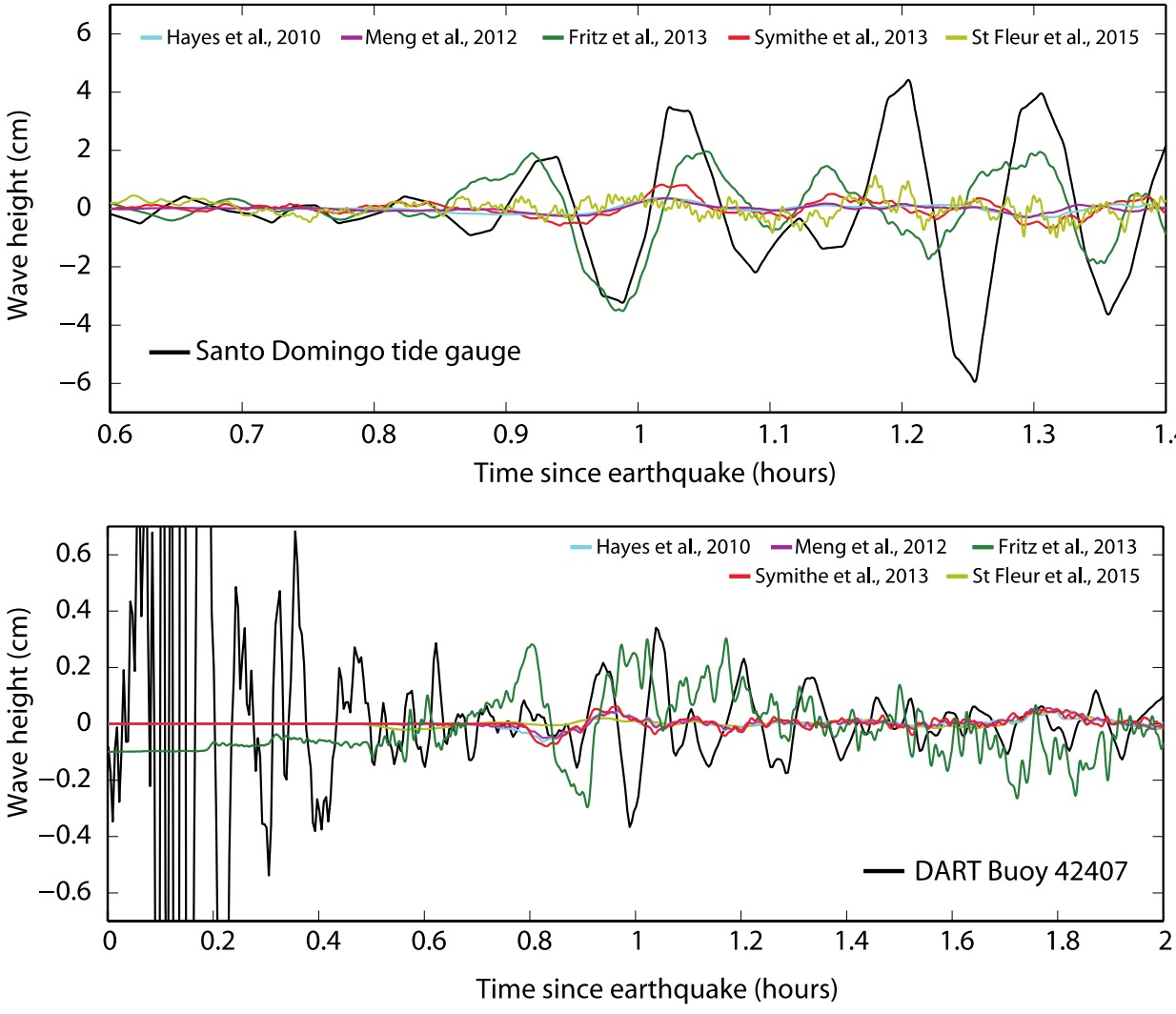

Figure 3: Comparison between observed and simulated water heights at the Santo Domingo tide gauge (top) and at DART buoy 42407 (bottom). Their locations are provided on Fig. 1. The high amplitude signal on the DART record before ~0.7 hours are seismic surface waves.

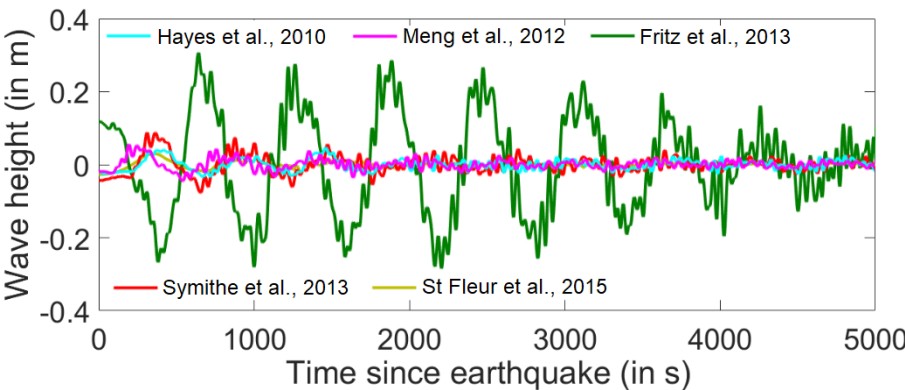

410 **Figure 4: Water height comparison at a synthetic gauge located in the center of the Bay of Jacmel at a 30 m water depth. Calculation on a bathymetric grid uses a 30 second resolution.**

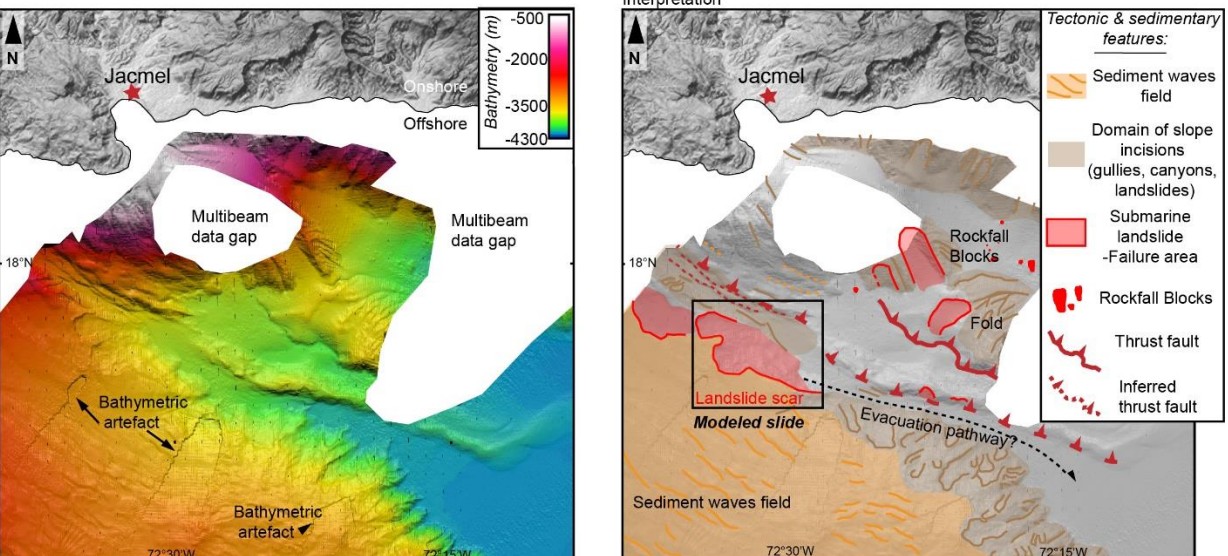

**Figure 5: Detailed bathymetry 20 km due south of Jacmel (Fig. 1) where the landslide scar discussed in the text has been identified. Left: raw observations. Right: geological interpretation. Swath bathymetric data are from Haiti-sis cruise (http://dx.doi.org/10.17600/12010070). White area corresponds to missing bathymetry of the boat campaign (Leroy et al., 2015).**

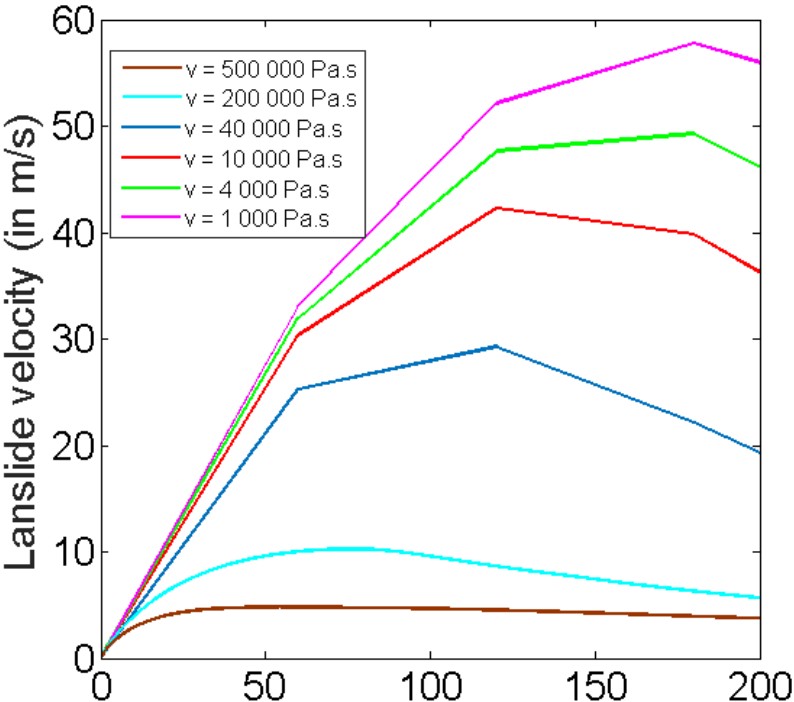

**Figure 6: Landslide velocity as a function of time for the dynamic viscosities $\nu$ tested here.**

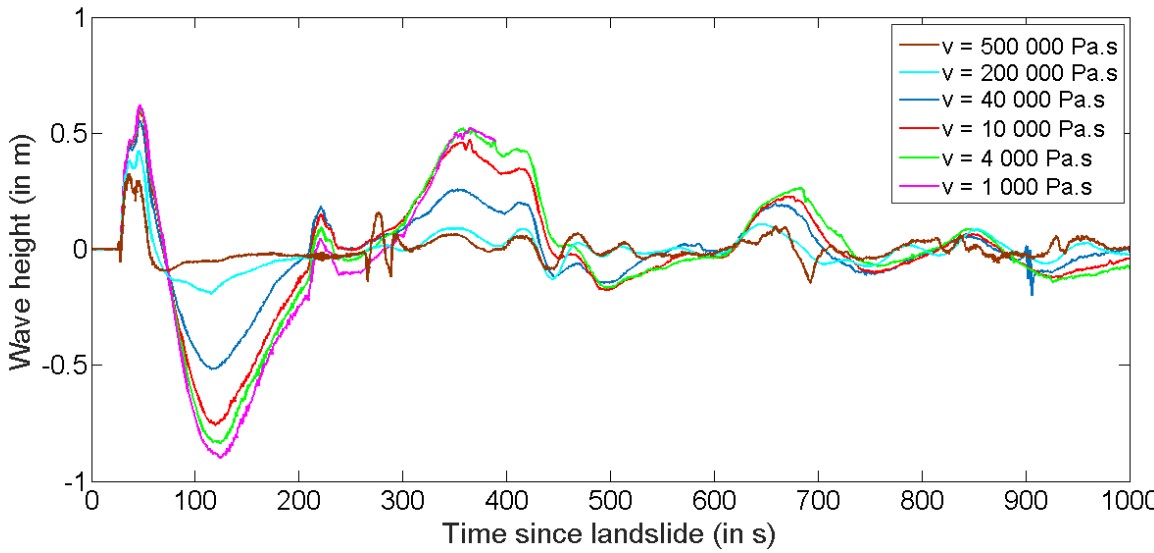

**Figure 7: Time series calculated at the synthetic gauge G$_1$ as a function of time for the dynamic viscosities $\nu$ tested here.**

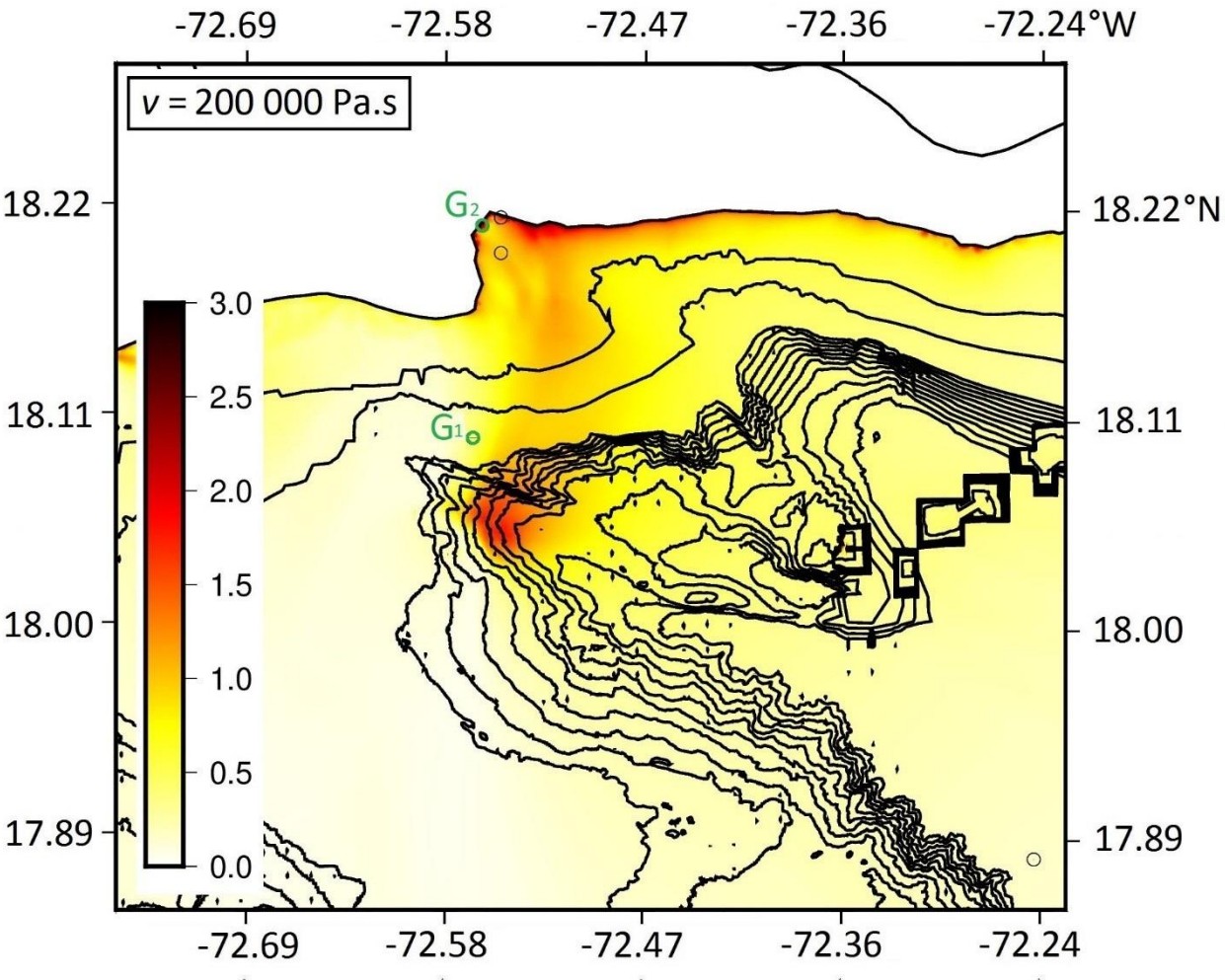


**Figure 8: Maximum modeled wave height using a viscosity $\nu = 2{\times}10^5$ Pa.s for the sliding material.**

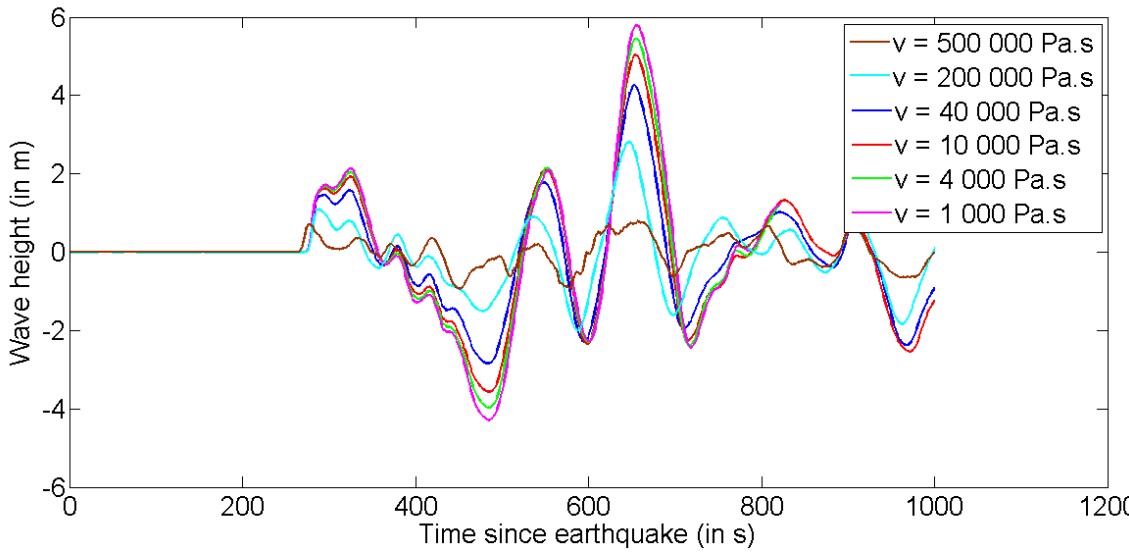

**Figure 9: Time series calculated at synthetic gauge G₂ located on the coast of the Bay of Jacmel (green circle in Fig. 8). This calculation uses a bathymetric grid with 25 m resolution.**

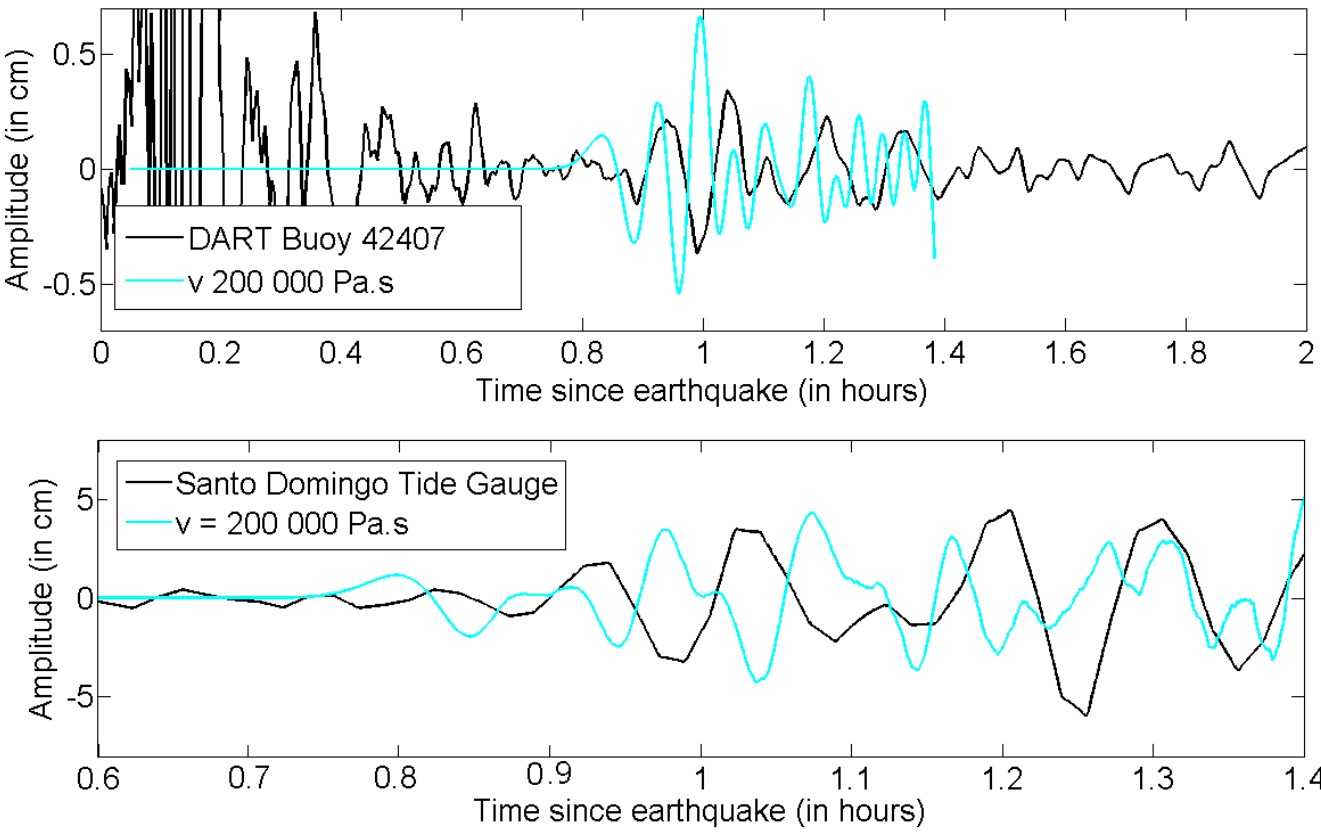

**Figure 10: Comparison between observed and simulated water heights at the Santo Domingo tide gauge (top) and at DART buoy 42407 (bottom) for a dynamic viscosity $\nu = 2 \times 10^5$ Pa.s. High amplitudes observed before 0.6 h is due to the earthquake detected by the DART Buoy.**

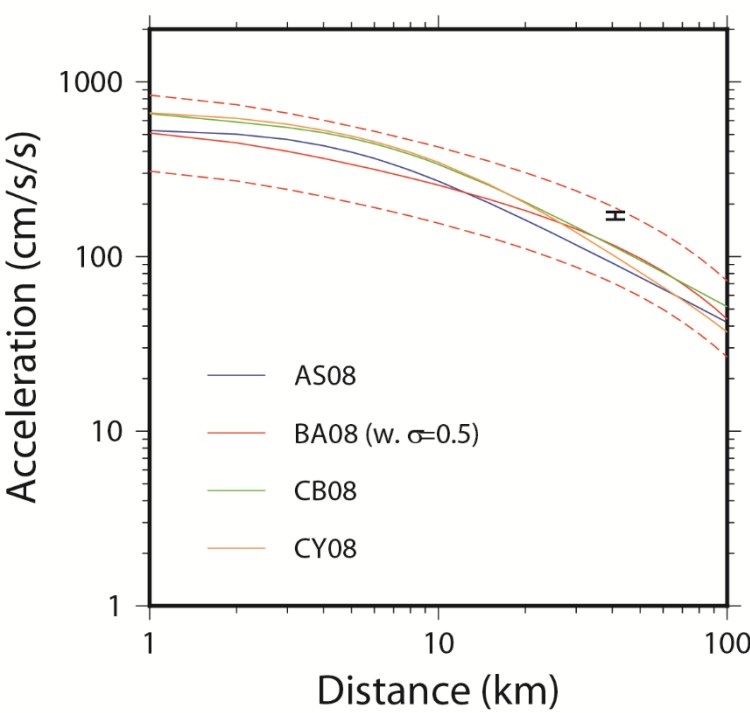

**Figure 11: Estimated horizontal peak ground accelerations due to an earthquake of M$_w$7.1 from four ground motion prediction equations: Abrahamson and Silva (2008; blue line), Boore and Atkinson (2008; solid red line with ±one standard deviation in dashed lines), Campbell and Bozorgnia (2008; yellow line) and Chiou and Youngs (2008; green line). The error bar symbol shows more accurate estimates derived from hybrid broadband ground acceleration calculation at the site of interest, as proposed in Douilly et al. (2017). This combines the deterministic low-frequency part from the dynamic rupture simulation of Douilly et al. (2015) with the stochastic high frequency one calculated through the Specific Barrier Model (Papageorgiou and Aki, 1983) for a crossover frequency of 1 Hz.**
