# Peer review of "Deep Submarine Landslide Contribution to the 2010 Haiti Earthquake Tsunami"

_Natural Hazards and Earth System Sciences, 2019_

## Referee Comment (RC1) · Anonymous Referee #1 · 30 Jan 2020

Poupardin et al. investigate the observation of 2 m tsunami run-up height at a distance of about 100 km away from the epicenter of the Haiti, Mw 7.0 strike slip earthquake, and the tide gauge signal recorded by the DART buoy 42407, about 600 km away from that epicenter.

The authors ". . . demonstrate that these observations require a secondary source, most likely a submarine landslide" and suggest a probable candidate off the southern coast of Haiti, about 30 km away from the epicenter. They conduct detailed modeling and are able to explain fairly well the unique observations.

The added value of this project is the better understanding of the potential of strike-slip earthquakes to generate tsunamis via submarine landslides.

[Figure]

This is an important paper, it is well structured, and points towards tsunamis that follow strike-slip earthquakes, a mechanism sometimes overlooked that deserves focused attention.

There are several comments I would like to suggest, mostly technical, and hope they will improve the manuscript. Overall, I do recommend the publishing of this work.

Comments

Line 12 and Figure 1: The magnitude in the text is Mw 7, and in Fig. 1 is Mw 7.1. . .

Line 48: ". . ., but the Haiti earthquake did not exhibit such a slow thrust rupture". Please give a reference.

Line 77: ". . . All finite-source models show that rupture occurred. . .": I would suggest the adding of a table of source parameters as well as 'beach balls' (fault plane solutions) in order to show the differences among the given models. This will also enable a better understanding of the various patterns of the coseismic seafloor deformation exhibited in Fig. 2.

Lines 98 – 99 and/or in the discussion: ". . . An offshore landslide comes to mind, as they are well known to trigger tsunamis even at large distances . . ." (also line 147): It would be interesting to compare the Haiti 'earthquake-submarine landslide-tsunami' sequence with the M-D relationships proposed by Salamon and Di-Manna (2019).

Lines 204-205: "It has been noted that tsunamis in strike-slip tectonic regimes were more frequent than expected. . .": This notion should be further elaborated, for that there are several modes of tsunami generation in strike-slip environments. For example, tsunamis due to: coseismic deformation (e.g. Imamura et al., 1995; Frucht et al., 2019); subaerial landslides; submarine landslides (e.g. Hoffmann et al., 2014); and all these three modes are relevant for both on-land and offshore sesimogenic sources.

Figures

Fig. 1: Should be Santo Domingo on the map?

Geographical coordinates are not easy to recognize.

Fig. 2: Abbreviations mentioned in the caption do not appear on the maps. . ..

Fig. 5: Please note whether the white areas are land or sea (although intuitively it looks like unmapped seafloor areas). Left side map: The blue color is missing from the bathymetry scale. Right side map: Legend font size is too small.

Fig. 8: Please round the coordinates' numbers, no need for four digits after the decimal point, one or two is enough.

Fig. 10: Please note which of the models is simulated here; The DART buoy is the upper diagram; Repeat explanation of the high amplitude signal of the DART buoy before ∼0.7 hour.

Fig. 11: Campbell and Bozorgnia state that their 2014 work supersedes their previous 2008 publication. Would this make any change in here? Is it important to note 'Mw 7.09' or Mw 7.1 is enough?

References (not mentioned in the text)

Campbell and Bozorgnia (2014) Earthquake Spectra, Volume 30, No. 3, pages 1087–1115.

Frucht, E., Salamon, A., Gal, E., Ginat, H., Grigorovitch, M., Shem Tov, R., Ward, S., (2019). A Fresh View of the Tsunami Generated by the Dead Sea Transform, 1995 Mw 7.2 Nuweiba Earthquake, along the Gulf of Elat-Aqaba. Seismological Research Letters, 90 (4): 1483-1493.

Hoffmann, G., Al-Yahyai, S., Naeem, G., Kociok, M., Grützner, C., 2014. An Indian Ocean tsunami triggered remotely by an onshore earthquake in Balochistan, Pakistan. Geology 42 (10), 883–886.

Imamura, F., Gica, E., Takahashi, T., Shuto, N., 1995. Numerical Simulation of the 1992 Flores Tsunami: Interpretation of Tsunami Phenomena in Northeastern Flores Island and Damage at Babi Island. Pure and Applied Geophysics 144, 555–568.

Salamon, A. and Di Manna, P., (2019). Empirical constraints on magnitude-distance relationships for seismically induced submarine tsunamigenic landslides. Earth-Science Reviews, 191: 66–92.

---

## Referee Comment (RC2) · Frederic Dias (Referee) · 1 Feb 2020

Frederic Dias (Referee)

frederic.dias@ucd.ie

This is a very good discussion paper on the 2010 Haiti earthquake tsunami. An additional landslide would explain some discrepancies between observations and simulations.

Specific comments: only two. (i) A little bit more should be said about the numerical model that solves the Boussinesq equations. Is there a reference? (ii) It would be interesting to perform a sensitivity study on the landslide parameters in the spirit of the analysis performed by Salmanidou et al. (2017) Statistical emulation of landslide-induced tsunamis at the Rockall Bank, NE Atlantic. Proc. R. Soc. Lond. A 473, 20170026.

[Figure]

Technical corrections: only two minor ones. (i) the authors sometimes use words, sometimes numbers to describe distances or whatever quantities; (ii) in Murphy & O'Brien, replace "accel- eration" by "acceleration"
* * *

---

## Referee Comment (RC3) · Anonymous Referee #3 · 6 Feb 2020

General Comments:

This is a well-written and potentially very impactful paper arguing for a submarine land-slide origin for a tsunami that struck the south coast of Haiti, relatively distant from the epicenter, after the 2010 earthquake. The authors have identified a likely landslide source and tested whether proposed finite-source models for the earthquake can account for the magnitude and timing of the tsunami as observed at a tide gauge and DART buoy. They compare these results to a model based on a tsunami triggered by the submarine landslide they have identified and argue that with realistic parameters, the submarine landslide model best explains the observed tsunami. The authors make a convincing case that a submarine landslide is indeed the most likely cause, and place their results in context of increasing recognition of the tsunami hazard posed by submarine landslide triggered by strike-slip earthquakes in particular. Their methodology could be beneficial in improving tsunami hazard models in earthquake-prone coastal regions where tsunami hazard models have not been accurately formulated. I recommend publication, and what questions and suggestions I have are mostly minor, although I think readers may have some of the same questions as I do about some of the model results and the authors' assumed parameters for the landslide. Since this manuscript is quite short and to-the-point, I think the authors should be able to clarify some of these points without it losing its punch. Additionally, the geography for some of the figures is a bit unclear (I had to google to figure out where Jacmel is) and should be made explicit.

Specific comments:

Line 112 – Is the feature possibly recognizable using pre-earthquake bathymetry? Such a large landslide should be recognizable in pretty coarse bathymetry, right? Based on Fig. 5, it would seem like the GEBCO 1 arc minute bathymetry available from the pre-seismic might be enough to resolve something. I'm convinced that the recognized slide is a likely source but pre-seismic bathymetric data, even at a low resolution could make it obvious that it's the smoking gun. Or, possibly, could it have been picked up on seismometers?

Line 131 – Should the resulting tsunami be similar if the landslide behaved more as a rigid sliding block than a viscous flow? It's difficult to tell from Fig. 5 whether the deposit is fairly coherent.

Line 139 – Was the tsunami reported anywhere at the southern end of the Caribbean?

Line 144 – Could this also be explained by a more coherent landslide block?

Figure 1 – Where is the Bay of Jacmel exactly?

Figure 2 – It would be nice to have a box on Fig. 1 showing this location.

Figure 3 – Why does the Fritz model show better temporal alignment in the early waves

than does the secondary source model? Is this purely coincidental? It's strange that the first few waves correlate so well with the Fritz model at the Santo Domingo gauge, whereas the landslide model doesn't seem to describe the arrival time quite as well.

Figures 3&10 – It would be helpful when comparing these to have the same X axes and the same order of tide gauge and DART buoy data.

Figure 5 – You have the callback to fig. 1 here, but I don't see fig. 5 outlined there.

Figure 6 – What would an even higher viscosity look like? Why choose 2e5 as a best-fit value without showing what a more viscous slide would produce?

Figure 10 – What happens after 1.5 hours at the Santo Domingo gauge? Why does the simulation build to its maximum amplitude at the very end? I don't know whether this actually matters, but I'd be curious to see what happens if the model is allowed to run a bit longer.

Technical corrections:

Line 115 – In this paragraph you switch between present and past tense frequently, try to keep the tense consistent.

Line 117 – should be "It consists of", also don't need the hyphen between filling and in.

Line 152 – Terrestrial instead of "on-land"

Line 156 – Capitalize "Holocene"

Line 184 – "associated with" instead of "associated to"

---

## Author Comment (AC1) · 30 Mar 2020

Review 1

Line 12 and Figure 1: The magnitude in the text is Mw 7, and in Fig. 1 is Mw 7.1...

OK. We changed all magnitudes in the text into Mw 7.1.

Line48: "...,but the Haiti earthquake did not exhibit such as low thrust rupture". Please give a reference.

Mercier de Lépinay et al. (GRL 2011, now cited) estimate rupture velocity at 2.6 km/s, a normal, not particularly slow, rupture velocity.

Line 77: "... All finite-source models show that rupture occurred...": I would suggest the adding of a table of source parameters as well as 'beach balls' (fault plane solutions) in order to show the differences among the given models. This will also enable a better understanding of the various patterns of the coseismic seafloor deformation exhibited in Fig. 2.

Because of complex ruptures described by several authors, beach ball cannot be used to represent the earthquake easily.

Lines 98 – 99 and/or in the discussion: "... An offshore landslide comes to mind, as they are well known to trigger tsunamis even at large distances ..." (also line 147): It would be interesting to compare the Haiti 'earthquake-submarine landslide-tsunami' sequence with the M-D relationships proposed by Salamon and Di-Manna (2019).

Indeed, following the relation given by Salamon and Dimanna, log(Re) = −0.87 + 0.45 Mw = 2.325 and then Re=211 km which is much larger than the distance between the earthquake epicenter and the landslide (around 70 km). We added a comment in the text.

Lines 204-205: "It has been noted that tsunamis in strike-slip tectonic regimes were more frequent than expected...": This notion should be further elaborated, for that there are several modes of tsunami generation in strike-slip environments. For example, tsunamis due to: coseismic deformation (e.g. Imamura et al., 1995; Frucht et al., 2019); subaerial landslides; submarine landslides (e.g. Hoffmann et al., 2014); and all these three modes are relevant for both on-land and offshore seismogenic sources.

OK. We made a distinction in the text.

Figures

Fig. 1: Should be Santo Domingo on the map? Geographical coordinates are not easy to recognize.

Santo Domingo is on the map of Figure 1 under the label "Santo Domingo tide gauge".

Fig. 2: Abbreviations mentioned in the caption do not appear on the maps....

OK. abbreviations have been added in figure 2 maps.

Fig. 5: Please note whether the white areas are land or sea (although intuitively it looks like unmapped seafloor areas). Left side map: The blue color is missing from the bathymetry scale. Right side map: Legend font size is too small.

Ok we now explain the white color signification in the legend. Figure 5 has been corrected.

Fig. 8: Please round the coordinates' numbers, none ed for four digits after the decimal point, one or two is enough.

OK. We replaced longitude and latitude by removing two digits.

Fig. 10: Please note which of the models is simulated here; The DART buoy is the upper diagram; Repeat explanation of the high amplitude signal of the DART buoy before~0.7 hour.

OK. We added a comment in the legend of Figure 10.

Fig. 11: Campbell and Bozorgnia state that their 2014 work supersedes their previous 2008 publication. Would this make any change in here? Is it important to note 'Mw 7.09' or Mw 7.1 is enough?

We used the original version of Campbell and Bozorgnia (2008) (not the updated version of 2014), as we wanted a brief order of the ground motion estimations for the given simple configurations.

OK. We modified the magnitude.

We added the mentioned references in the text.

---

## Author Comment (AC2) · 30 Mar 2020

Specific comments: only two. (i) A little bit more should be said about the numerical model that solves the Boussinesq equations. Is there a reference? (ii) It would be interesting to perform a sensitivity study on the landslide parameters in the spirit of the analysis performed by Salmanidou et al. (2017) Statistical emulation of landslide induced tsunamis at the Rockall Bank, NE Atlantic. Proc. R. Soc. Lond. A 473, 20170026.

The model is described in Poupardin et al. (GJI). We added this reference in the paper.

In our mind, this sensitivity study is beyond the scope of this paper. Indeed, we considered a viscous flow, the volume is fixed by the ground scar, and we varied the viscosity of the landslide.

Technical corrections: only two minor ones. (i) the authors sometimes use words, sometimes numbers to describe distances or whatever quantities; (ii) in Murphy & O'Brien, replace "accel- eration" by "acceleration"

OK. We corrected some numbers in the text and the reference.

---

## Author Comment (AC3) · 30 Mar 2020

Specific comments:

Line 112 – Is the feature possibly recognizable using pre-earthquake bathymetry? Such a large landslide should be recognizable in pretty coarse bathymetry, right? Based on Fig. 5, it would seem like the GEBCO 1 arc minute bathymetry available from the pre-seismic might be enough to resolve something. I'm convinced that the recognized slide is a likely source but pre-seismic bathymetric data, even at a low resolution could make it obvious that it's the smoking gun. Or, possibly, could it have been picked up on seismometers?

By using GEBCO data at 15 s and 1 min, we got:

[Figure]

GEBCO data give us a water depth between 3800 and 3900 m near the landslide which seems consistent with oceanographic campaign data acquired in 2012. We do not see anything around 3500 m that could be assimilated to the ground scar identified in the 2012 data. Then two possibilities may be considered: the first one is that the landslide occurred after the 2010 Haiti earthquake; the second one is that GEBCO data are not precise enough to identify a ground scar with a thickness of 100 m.

We added some comments in text to mention that we used GEBCO data to search a potential ground scar.

Line 131 – Should the resulting tsunami be similar if the landslide behaved more as a rigid sliding block than a viscous flow? It's difficult to tell from Fig. 5 whether the deposit is fairly coherent.

As sediment profiles are not available in the area we cannot conclude on the soil nature which impacts the landslide form. More precise sedimentologic studies must be led to conclude on the landslide age, mechanics properties and lithology.

Line 139 – Was the tsunami reported anywhere at the southern end of the Caribbean?

To our knowledge, the tsunami was not observed anywhere else but according to Fritz simulation (using the NEIC source), Cuban and Jamaican coasts could have been hit by waves of 2 cm high.

Line 144 – Could this also be explained by a more coherent landslide block?

First, it has be noted that "a rigid-like motion for a landslide is realistic only for ideal conditions where the sliding mass has an ideal shape and moves on a smooth topography" '(Yavari-Ramshe and Ataie-Ashtiani, 2016). In our case, the complex slope does not allow to deal with the slump of a rigid block.

Considering a regular slope, Lovholt et al. (2015) compare water waves generated by a rigid slump and a deformable landslide (Fig. 4 in Lovholt et al., 2015). They show that the landslide deformation has only a weak influence on waveforms. According to the analytical analysis of Glimsdal et al. (2013), the spectrum of water waves is mainly influenced by the length of the landslide.

Nevertheless, according to the majority of authors, the energy transfer from a landslide into the water surface is larger for rigid blocks (Yavari-Ramshe and Ataie-Ashtiani, 2016) and the rigid-block assumption may result in overestimating wave amplitudes up to 30%. Finally, the review of Yavari-Ramshe and Ataie-Ashtiani emphasizes on the necessity of further investigations on the effects of landslide rigidity and on energy transferring.

Yavari-Ramshe and Ataie-Ashtiani (2016). Numerical modeling of subaerial and submarine landslide-generated tsunami waves—recent advances and future challenges. Landslides 13:1325–1368, DOI 10.1007/s10346-016-0734-2.
Glimsdal S, Pedersen GK, Harbitz CB, Løvholt F (2013) Dispersion of tsunamis: does it really matter? Nat Hazards Earth Syst Sci 13:1507–1526. doi:10.5194/nhess-13-1507-2013
Løvholt, Finn & Pedersen, G & Harbitz, Carl & Glimsdal, Sylfest & Kim, Jihwan. (2015). On the characteristics of landslide tsunamis. Philosophical transactions. Series A, Mathematical, physical, and engineering sciences. 373. 10.1098/rsta.2014.0376.

Figure 1 – Where is the Bay of Jacmel exactly?

Label now added to Figure 1.

Figure 2 – It would be nice to have a box on Fig. 1 showing this location.

It has been added.

Figure 3 – Why does the Fritz model show better temporal alignment in the early waves than does the secondary source model? Is this purely coincidental? It's strange that the first few waves correlate so well with the Fritz model at the Santo Domingo gauge, whereas the landslide model doesn't seem to describe the arrival time quite as well.

A slight delay of 8 minutes exists between observations and simulations (Fig. 10). This could be explained if the collapse was triggered some minutes after the earthquake but it is not consistent with our simulations in the bay of Jacmel (Fig. 9).

[Figure]

A comment was added in the text.

Figures 3&10 – It would be helpful when comparing these to have the same X axes and the same order of tide gauge and DART buoy data.

OK. We corrected the X axis in Fig. 10.

Figure 5 – You have the callback to fig. 1 here, but I don't see fig. 5 outlined there.

The outline of Figure 1 is now shown on Figure 1.

Figure6–What would an even higher viscosity look like? Why choose 2e5 as a best-fit value without showing what a more viscous slide would produce?

We chose this value because it allows us to reproduce the 3 m water height observed in Jacmel Bay.

We added results for a viscous flow of 500 000 Pa.s viscosity in figures manuscript.

Figure 10 – What happens after 1.5 hours at the Santo Domingo gauge? Why does the simulation build to its maximum amplitude at the very end? I don't know whether this actually matters, but I'd be curious to see what happens if the model is allowed to run a bit longer.

The one-way coupling between daughter and mother grids induces numerical oscillations in the daughter grid after 1,5 hours of propagation. These oscillations are generated at the daughter grid boundaries and are due to differences of depth values between grids. The following comment has been added in the legend of Figure 10:

"Beyond 1.5 hours of propagation, the simulated water waves are affected by numerical oscillations due to the one-way coupling between grids and are not significant"

Technical corrections:

Line 115 – In this paragraph you switch between present and past tense frequently, try to keep the tense consistent.

OK we corrected this paragraph by using present tense.

Line 117 – should be "It consists of", also don't need the hyphen between filling and in.

Ok. We corrected it in the text.

Line 152 – Terrestrial instead of "on-land"

Ok. We corrected it in the text.

Line 156 – Capitalize "Holocene"

Ok. We corrected it in the text.

Line 184 – "associated with" instead of "associated to"

Ok. We corrected it in the text.

---

## Author Response (AR1)

We thank all reviewer and editor for their constructive comments which improve this manuscript on the ability of large landslide to generate tsunami in Caribbean Seas.

---

## Author Response (AR2)

Dear editor,

We thank you for your last comments.

As requested, we modify the conclusion by moving a paragraph in the discussion part.

Furthermore, we add a paragraph to discuss on a new publication by tan Brink recently published.

Kind regards

Adrien Poupardin and co-authors